# Peripheral Neuropathy Potentially Associated to Poly (ADP-Ribose) Polymerase Inhibitors: An Analysis of the Eudravigilance Database

Mafalda Jesus [1,2], António Cabral [1,3], Cristina Monteiro [1,2,4], Ana Paula Duarte [1,2,4] and Manuel Morgado [1,2,5,*]

1 Health Sciences Faculty, University of Beira Interior (FCS-UBI), 6200-506 Covilhã, Portugal; mafalda.jesus@ubi.pt (M.J.); antonio.clopes@ulsguarda.min-saude.pt (A.C.); csjmonte@ubi.pt (C.M.); apcd@ubi.pt (A.P.D.)
2 Health Sciences Research Center, University of Beira Interior (CICS-UBI), 6200-506 Covilhã, Portugal
3 Pharmaceutical Services of Local Healthcare Unit of Guarda, 6300-749 Guarda, Portugal
4 UFBI—Pharmacovigilance Unit of Beira Interior, University of Beira Interior, 6200-506 Covilhã, Portugal
5 Pharmaceutical Services of University Hospital Center of Cova da Beira, 6200-251 Covilhã, Portugal
* Correspondence: mmorgado@fcsaude.ubi.pt

**Abstract:** Poly (ADP-Ribose) polymerase inhibitors (PARPi) have emerged as a targeted therapy in cancer treatment with promising results in various types of cancer. This work aims to investigate the profile of adverse drug reactions (ADRs) associated with PARPi through the reports provided by the Eudravigilance (EV) database. We also intend to analyze the potential association of peripheral neuropathy to PARPi. Data on individual case safety reports (ICSRs) were obtained by accessing the European spontaneous reporting system via the EV website. A total of 12,762 ICSRs were collected from the EV database. Serious cases of nervous system disorders were analyzed providing strong evidence that peripheral neuropathy was reported in a higher frequency in patients treated with niraparib. Most cases reported a not recovered/not resolved outcome and involved drug withdrawal. However, several studies suggest that PARPi attenuate chemotherapy-induced painful neuropathy. Unexpected ADRs such as peripheral neuropathy may also occur, mostly in patients taking niraparib. Further pharmacovigilance studies should be conducted in this area to clarify with more precision the toxicity profile of these drugs.

**Keywords:** adverse drug reactions; PARP inhibitors; Eudravigilance database; nervous and system disorders; peripheral neuropathy

## 1. Introduction

Poly (ADP-Ribose) polymerase inhibitors (PARPi) are a type of targeted anti-cancer therapy. Several theories have emerged to describe the precise mechanism of action by which PARPi induce their anti-cancer activity, although a consensus is yet to be reached [1]. In this context, two of the most described methods rely on inhibition of the PARP enzyme and on the "PARP trapping" concept, PARP1 being the main target [2,3]. PARP1 are considered the most abundant isoform and are recognized by their role in DNA repair processes. Recent evidence highlights their role in several cell processes, ranging from cell proliferation to cell death [4]. Other isoforms such as PARP2, PARP3, and PARP5 are described in the literature. PARP2, PARP3, and PARP5 isoenzymes share some of the physiological functions of PARP1 [5–7].

Over the last decade, the U.S. Food and Drug Agency (FDA) and European Medicines Agency (EMA) have already approved four PARPi, namely: olaparib, niraparib, rucaparib and talazoparib. Olaparib was the first PARPi to be developed, approved in 2014 by FDA and EMA [8]. Briefly, the approved clinical indications focus on the maintenance therapy

as a first line in ovarian cancer and as a second line in recurrent ovarian cancer after platinum, recurrent metastatic ovarian and breast cancer, maintenance of metastatic pancreatic cancer after chemotherapy, and recurrent metastatic castration-resistant prostate cancer (mCRPC) [9–12]. Rucaparib was approved for the maintenance treatment of recurrent ovarian cancer as a second line and for recurrent metastatic ovarian cancer [13,14], followed by the approval of niraparib, with clinical indications for the treatment of ovarian cancer, similarly to olaparib [15,16]. Talazoparib was the last PARPi to be approved, in 2019, and is used for the treatment of recurrent, metastatic breast cancer [17,18].

Although targeted therapies are generally associated with fewer side effects, the literature describes several adverse drug reactions (ADRs) associated with PARPi [19]. A specific pattern of ADRs including fatigue, hematological, gastrointestinal, nervous, metabolism, and nutritional disorders can be found [20–23]. Indeed, fatigue is considered the most common ADRs seen in patients taking these drugs. According to the literature, 59–69% of patients had fatigue of any degree with olaparib, rucaparib, and niraparib [9,16,24]. In the phase III trials [the SOLO-1 trial (olaparib monotherapy 300 mg twice daily vs. placebo), the PAOLA-1 trial (olaparib 300 mg twice daily plus bevacizumab vs. bevacizumab monotherapy), and the PRIMA trial (niraparib monotherapy 300 mg qd vs. placebo)], all PARPi as FDA-approved maintenance therapy after first-line platinum therapy showed high rates of a number of common non-hematologic ADRs, such as nausea, vomiting, and fatigue/asthenia, which are generally low-grade and rarely lead to study drug discontinuation [25–27]. Among hematological adverse events, anemia is considered the most common. Grade 3 and 4 events are considered the most common cause of dose adjustment or drug discontinuation, particularly in patients taking niraparib [28]. Additionally, nausea and, to a lesser extent, vomiting are very common ADRs associated with PARPi, especially in patients with ovarian cancer. Supportive treatment including antiemetics are usually effective, avoiding dose interruption and dose reduction actions [29,30]. Elevated creatinine, liver enzymes and cholesterol were also mentioned as common investigational toxicities of patients using PARPi [31]. Other less common ADRs are reported, even though their frequency cannot be entirely ignored. For example, nervous symptoms such as headache and dizziness are often related with patients taking olaparib, niraparib, and rucaparib [31]. Dysgeusia is more common with olaparib and rucaparib, with cardiac/cardiovascular disorders, particularly palpitations and hypertension, being more frequently associated with niraparib [19]. Additionally, cutaneous disturbs have been identified in patients taking olaparib, rucaparib, and niraparib. However, only the ARIEL3 trial mentioned these disturbs and, consequently, their association with rucaparib use [24]. Rare and delayed adverse effects should also be highlighted, such as myelodysplastic syndrome and acute myeloid leukemia [19]. In addition, a recent real-world pharmacovigilance study of FDA Adverse Event Reporting System (FAERS) described unexpected and new significant ADRs such as peripheral neuropathy involving niraparib [32]. In fact, it is well known that chemotherapeutic agents can induce peripheral neuropathy, manifesting in symptoms such as paresthesia, hyperalgesia, and allodynia. Chemotherapy-induced peripheral neuropathy can persist from months to years after chemotherapy completion, causing a negative influence on function and quality of life in cancer patients [33,34].

PARPi have shown their clinical relevance in the management of patients with various malignancies and an increase in their use is expected in following years. In addition, it is important to highlight that although PARPi share several adverse effects among them, differences can be identified due to variations in their poly-pharmacology and off-target effects. It should also be noted that most studies on the adverse effects of PARPi are currently based on data from clinical trials [35]. Thus, the present pharmacovigilance study aims to analyze the profile of suspected ADRs reported for olaparib, rucaparib, niraparib, and talazoparib in a real world setting through the analysis of the Eudravigilance (EV) data. In a more detailed way, we also intend to analyze the potential association of peripheral neuropathy with PARPi, a condition that has significant impact on cancer patients' lives.

## 2. Materials and Methods

### 2.1. Data Source

Data on individual case safety reports (ICSRs) were retrieved from the website of suspected ADRs of the EV database by accessing www.adrreports.eu (accessed on 8 March 2023). The EV is a system for managing and analyzing ICSRs of suspected ADRs related to medicines which have been authorized or are being studied in clinical trials in the European Economic Area (EEA) [36,37].

### 2.2. Individual Cases Safety Reports Selection and Descriptive Analysis

- In the EV database, by using the line listing function, we selected all ICSRs with a PARPi as suspected drug and reported from the date of marketing authorization granted by EMA for each PARPi to 1 March 2023. In this context, the marketing authorization dates were the following: 16 December 2014 for olaparib; 16 November 2017 for niraparib; 23 May 2018 for rucaparib; and 20 June 2019 for talazoparib. Information was collected on sex, age group, reporter group, geographic origin, outcome by reaction group (10 most reported ADRs were considered), and seriousness. According to the International Council on Harmonization E2D guidelines, a case is defined as serious if it results in death, is life threatening, requires or prolongs a hospitalization, results in disability/incapacity, determines a congenital anomaly/birth defect, or results in other medically important information [38].
- Following the previous steps, qualitative and quantitative analyses were performed for the least reported outcomes of ICSRs from 1 January 2022 to 31 December 2022. For the suspected drugs in study, the main reported conditions were highlighted, considering only serious cases. All suspected ADRs reports in which PARPi were not described as the only suspected drug were excluded.
- A more detailed analysis was also performed by selecting all ICSRs with peripheral neuropathy as a reported suspected reaction from 1 January 2022 to 31 December 2022. Only serious cases were considered and information was collected on sex, age group, outcome, seriousness criteria, action taken, number of nervous disorders per ICSR, number of concomitant medicines per ICSR, and the overall number of suspected ADRs reported. All suspected ADRs reports in which PARPi were not described as the only suspected drug were excluded.
- Categorical variables were described through their absolute and relative frequency by using Office® Excel® 365 software, Version 2208 (Microsoft Corporation, Redmond, WA, USA). Pearson's Chi-Square test was used to verify a possible relationship between the variables with a statistical significance level of 5% ($p < 0.05$). In this case, IBM SPSS statistics 28 (IBM, Armonk, NY, USA) was used.
- Each ICSR may include one or more suspected ADRs. ADRs included in each ICSR were analyzed according to the Medical Dictionary for Regulatory Activities (MedDRA). MedDRA is a rich and highly specific standardized medical terminology to facilitate international sharing of regulatory information for medical products used by humans (https://www.meddra.org, accessed on 8 March 2023). In this context, the suspected ADRs mentioned in each ICSR are grouped in accordance with System Organ Classes (SOC).

## 3. Results

### 3.1. Demographic Characteristics of ICSRs

A total of 12,762 ICSRs have been reported considering a PARPi as a suspected drug, since the date of marketing authorization granted by EMA (olaparib—16 December 2014; niraparib—16 November 2017; rucaparib—23 May 2018; and talazoparib—20 June 2019) until 1 March 2023. More precisely, 5659 ICSRs pertained to olaparib, 5639 to niraparib, 1295 to rucaparib, and 169 to talazoparib. Most cases were reported in female patients (N = 11,499, 90.1%) compared to male patients (N = 472, 3.7%). In this context, a statistically significant difference was found between sex and the PARPi used ($p < 0.00001$). A high

number of ICSRs were considered "Not specified" in terms of age group (N = 5474, 42.9%), followed by 3670 (28.8%) cases for the age group 18–64 years and 3455 (27%) cases for the age group 65–85 years. In addition, healthcare professionals reported many of the cases (N = 9921, 77.7%) and most ICSRs came from the Non-European Economic Area (N = 9120, 71.5%). Regarding the individual cases reported by SOC, "General disorders and administration site conditions" and "Investigations" were the most described with 4468 and 4292 ICSRs, respectively. Additionally, a high prevalence of "Blood and lymphatic disorders" and "Gastrointestinal disorders" was noticed. Concerning the seriousness of the reported cases, a high percentage was classified as a serious case (10,814, 84.7%) when compared to non-serious cases (N = 1948, 15.3%). In this sense, a statistically significant difference ($p < 0.00001$) was found between seriousness and the drugs in study. These results are presented in Table 1. Figure 1 presents the contribution of each studied category (gender, age group, reporter group, region, and seriousness) to the total of ICSRs analyzed.

**Table 1.** Demographic characteristics of ICSRs involving PARPi since the date of marketing authorization granted by EMA (olaparib—16 December 2014; niraparib—16 November 2017; rucaparib—23 May 2018; and talazoparib—20 June 2019) until 1 March 2023, according to the Eudravigilance database.

| | Individual Case Safety Reports (%) | | | | |
|---|---|---|---|---|---|
| | Olaparib N = 5659 | Niraparib N = 5639 | Rucaparib N = 1295 | Talazoparib N = 169 | Total N = 12,762 |
| Sex [a] | | | | | |
| Male | 382 (6.8) | 19 (0.4) | 63 (4.9) | 8 (4.7) | 472 (3.7) |
| Female | 5177 (91.4) | 4992 (88.5) | 1172 (90.5) | 158 (93.5) | 11,499 (90.1) |
| Not specified | 100 (1.8) | 628 (11.1) | 60 (4.6) | 3 (1.8) | 791 (6.2) |
| Age group | | | | | |
| Paedriatics (<18 years) | 6 (0.1) | 1 | 0 | 2 (1.2) | 9 (0.1) |
| Adult (18–64 years) | 1828 (32.3) | 1442 (25.6) | 292 (22.6) | 108 (63.9) | 3670 (28.8) |
| Elderly (65–85 years) | 1303 (23.0) | 1503 (26.7) | 616 (47.6) | 33 (19.5) | 3455 (27.0) |
| Very Eldery (>85 years) | 42 (0.8) | 78 (1.4) | 34 (2.6) | 0 | 154 (1.2) |
| Not Specified | 2480 (43.8) | 2615 (46.3) | 353 (27.2) | 26 (15.4) | 5474 (42.9) |
| Reporter group | | | | | |
| Health care professional | 4878 (86.2) | 3660 (64.9) | 1278 (98.7) | 105 (62.1) | 9921 (77.7) |
| Non-health care professional | 781 (13.8) | 1979 (35.1) | 17 (1.3) | 64 (37.9) | 2841 (22.3) |
| Region | | | | | |
| European Economic Area | 2184 (38.6) | 1018 (18.1) | 383 (29.6) | 57 (33.7) | 3642 (28.5) |
| Non-European Economic Area | 3475 (61.4) | 4621 (81.9) | 912 (70.4) | 112 (66.3) | 9120 (71.5) |
| Individual cases reported by system organ classes (SOC) [b] | | | | | |
| General disorders and administration site conditions | 1379 | 2463 | 592 | 34 | 4468 |
| Investigations | 1101 | 2733 | 417 | 41 | 4292 |
| Blood and lymphatic disorders | 2232 | 1632 | 246 | 85 | 4195 |
| Gastrointestinal disorders | 1276 | 2101 | 361 | 13 | 3751 |
| Neoplasms benign, malignant and unspecified (incl cysts and polyps) | 1384 | 1491 | 403 | 37 | 3315 |
| Injury, poisoning and procedural complications | 582 | 1332 | 183 | 20 | 2117 |
| Nervous system disorders | 448 | 1312 | 196 | 10 | 1966 |
| Respiratory, thoracic and mediastinal disorders | 482 | 782 | 118 | 12 | 1394 |
| Psychiatric disorders | 97 | 930 | 74 | 2 | 1103 |
| Skin and subcutaneous tissue disorders | 322 | 582 | 94 | 11 | 1009 |

**Table 1.** *Cont.*

| | Individual Case Safety Reports (%) | | | | |
| --- | --- | --- | --- | --- | --- |
| | Olaparib N = 5659 | Niraparib N = 5639 | Rucaparib N = 1295 | Talazoparib N = 169 | Total N = 12,762 |
| Number of individual cases [a] | | | | | |
| Serious | 4434 (78.4) | 5152 (91.4) | 1082 (83.6) | 146 (86.4) | 10,814 (84.7) |
| Non serious | 1225 (21.6) | 487 (8.6) | 213 (16.4) | 23 (13.6) | 1948 (15.3) |

[a] Pearson's Chi-Square test was used to verify a possible relationship between these variables with a statistical significance level of 5% ($p < 0.05$). [b] 10 most reported ADRs were analyzed.

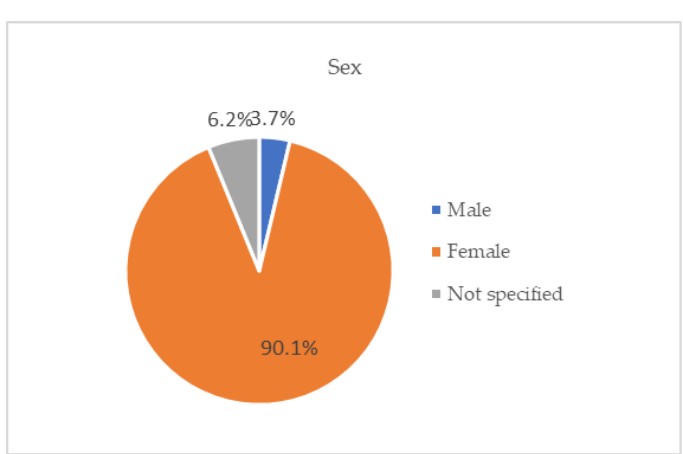

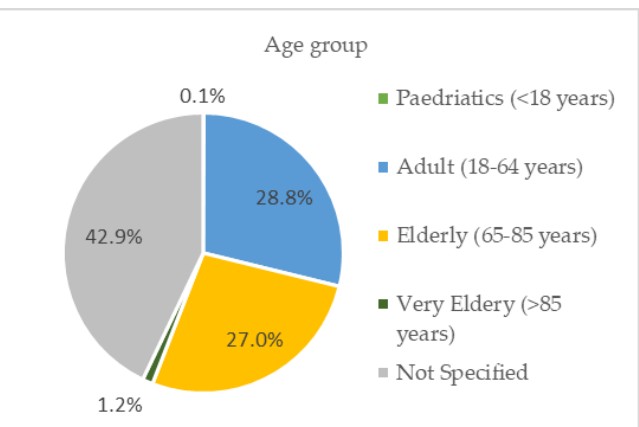

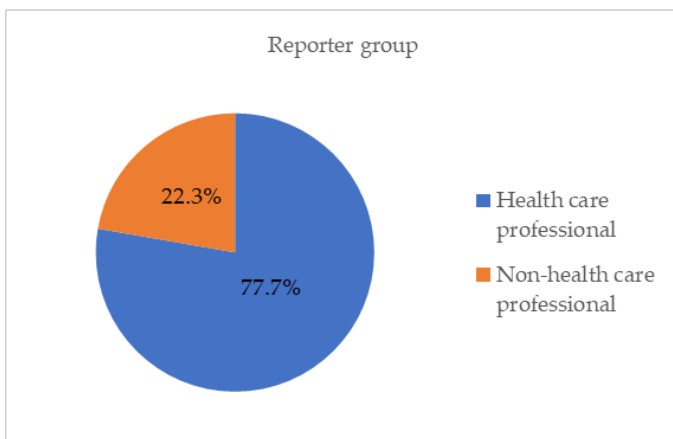

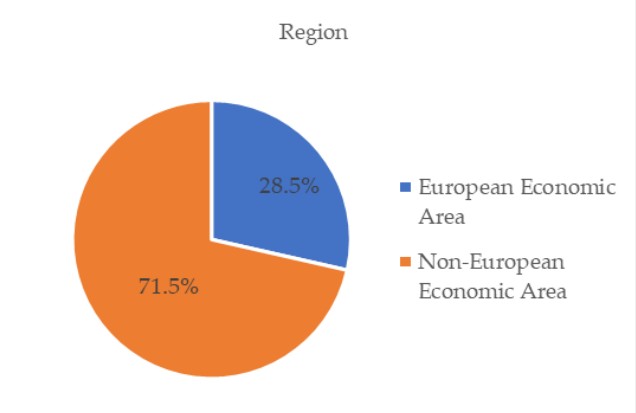

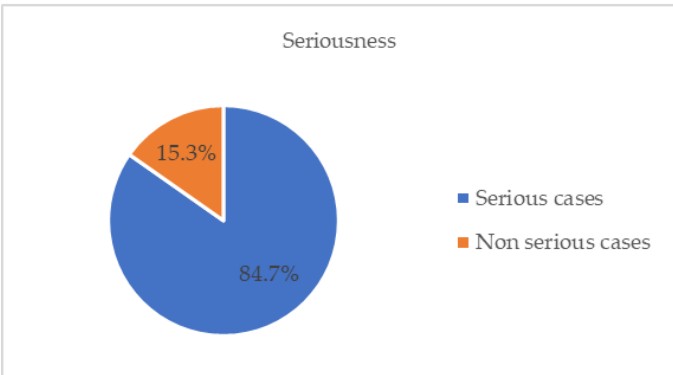

**Figure 1.** ICSRs having PARPi as suspect drugs sent through the EV database [since the date of marketing authorization granted by EMA (olaparib—16 December 2014; niraparib—16 November 2017; rucaparib—23 May 2018; and talazoparib—20 June 2019) until 1 March 2023].

*3.2. Least Reported SOCs*

The least reported SOCs presented in Table 1 were analyzed from 1 January 2022 to 31 December 2022. Considering the SOC "Respiratory, thoracic and mediastinal disorders", no statistically significant difference was found between the different PARPi (*p*-value = 0.25796). However, considering the other SOCs, "Injury, poisoning and procedural complications", "Nervous system disorders", "Psychiatric disorders", and "Skin and subcutaneous tissue disorders", a statistically significant difference was found between PARPi, concerning the SOCs mentioned. Regarding the SOC "Injury, poisoning and procedural complications", off-label use, product dose omission issue, product dose omission in error, contusion, and fall were the main safety issues reported considering all PARPi. Regarding the SOC "Nervous system disorders", headache, peripheral neuropathy, dizziness, taste disorder, and hypoaesthesia were the main reported symptoms considering all PARPi. These results, as well as the main safety issues reported for each mentioned SOC, are presented in Table 2. Figure 2 presents the distribution of the total serious ICSRs reported for each PARPi vs. serious ICSRs reported for some selected SOCs (least reported SOCs), only in the year of 2022 (from 1 January 2022 to 31 December 2022).

All columns highlighted represent the total number of serious ICSRs according to the SOC studied. All suspected ADRs reports in which PARPi were not described as the only suspected drug were excluded.

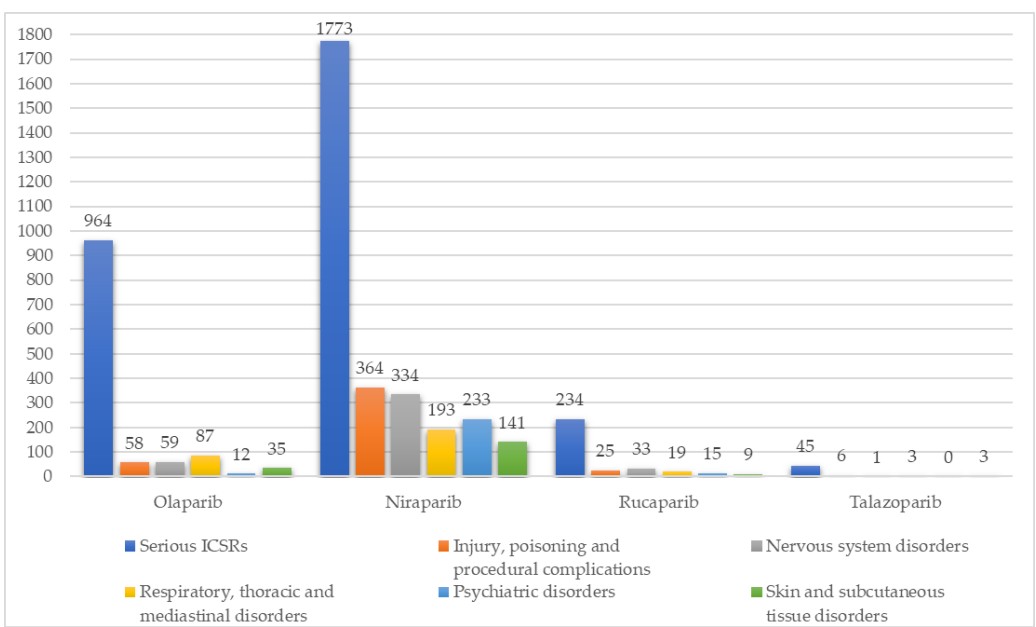

**Figure 2.** Distribution of the total serious ICSRs reported for each PARPi vs. serious ICSRs reported for some selected SOCs (least reported SOCs), from 1 January 2022 to 31 December 2022.

**Table 2.** Qualitative and quantitative analysis of the least reported SOCs, only in the year of 2022 (from 1 January 2022 to 31 December 2022). For each mentioned SOC, only the most reported safety issues were included, considering all PARPi.

| | | Individual Case Safety Reports (%) | | | | | | | | |
|---|---|---|---|---|---|---|---|---|---|---|
| | | Olaparib Total (N = 964) | | Niraparib Total (N = 1773) | | Rucaparib Total (N = 234) | | Talazoparib Total (N = 45) | | *p*-Value [a] |
| **SOC** | | | | | | | | | | |
| Injury, poisoning and procedural complications | Off label use | | 32 (55.2) | | 160 (43.4) | | 0 | | 3 (50.0) | |
| | Product dose omission issue | | 10 (17.2) | | 143 (39.3) | | 13 (52.0) | | 1 (16.7) | |
| | Product dose omission in error | 58 (6.0) | 0 | 364 (20.5) | 31 (8.5) | 25 (10.7) | 0 | 6 (13.3) | 0 | <0.00001 |
| | Contusion | | 3 (5.2) | | 23 (6.3) | | 0 | | 0 | |
| | Fall | | 2 (3.4) | | 14 (3.8) | | 4 (16.0) | | 1 (16.7) | |
| Nervous system disorders | Headache | | 8 (13.6) | | 133 (39.8) | | 9 (27.3) | | 0 | |
| | Peripheral neuropathy | | 2 (3.4) | | 87 (26.0) | | 11 (33.3) | | 0 | |
| | Dizziness | 59 (6.1) | 11 (18.6) | 334 (18.8) | 82 (24.6) | 33 (14.1) | 2 (6.1) | 1 (2.2) | 0 | <0.00001 |
| | Taste disorder | | 8 (13.6) | | 19 (5.7) | | 1 (3.0) | | 0 | |
| | Hypoaesthesia | | 2 (3.4) | | 21 (6.3) | | 3 (9.1) | | 0 | |
| Respiratory, thoracic and mediastinal disorders | Dyspnoea | | 9 (10.3) | | 79 (41.0) | | 9 (47.4) | | 2 (66.7) | |
| | Interstitial lung disease | | 43 (49.4) | | 8 (4.1) | | 0 | | 0 | |
| | Cough | 87 (9.0) | 3 (3.4) | 193 (10.9) | 37 (19.2) | 19 (8.1) | 2 (10.5) | 3 (6.7) | 0 | 0.25796 |
| | Oropharyngeal pain | | 2 (2.3) | | 20 (10.4) | | 0 | | 0 | |
| | Epistaxis | | 1 (1.1) | | 20 (10.4) | | 0 | | 0 | |
| Psychiatric disorders | Insomnia | | 1 (8.3) | | 155 (66.5) | | 7 (46.7) | | 0 | |
| | Anxiety | | 2 (16.7) | | 35 (15.0) | | 2 (13.3) | | 0 | |
| | Sleep disorder | 12 (1.2) | 0 | 233 (13.1) | 19 (8.2) | 15 (6.4) | 0 | 0 | 0 | <0.00001 |
| | Depression | | 1 (8.3) | | 14 (6.0) | | 4 (26.7) | | 0 | |
| | Confusional state | | 1 (8.3) | | 9 (3.9) | | 2 (13.3) | | 0 | |
| Skin and subcutaneous tissue disorders | Pruritus | | 5 (14.3) | | 35 (24.8) | | 5 (55.6) | | 0 | |
| | Rash | | 6 (17.1) | | 21 (14.9) | | 3 (33.3) | | 0 | |
| | Alopecia | 35 (3.6) | 3 (8.6) | 141 (8.0) | 11 (7.8) | 9 (3.8) | 1 (11.1) | 3 (6.7) | 1 (33.3) | 0.000053 |
| | Photosensitivity reaction | | 0 | | 21 (14.9) | | 1 (11.1) | | 0 | |
| | Erythema | | 5 (14.3) | | 9 (6.4) | | 0 | | 0 | |

[a] Pearson's Chi-Square test was used to verify a possible relationship between these variables with a statistical significance level of 5% (*p* < 0.05).

### 3.3. Analysis of ICSRs—Peripheral Neuropathy

A detailed analysis of peripheral neuropathy serious cases was performed from 1 January 2022 to 31 December 2022 for the inhibitors olaparib, niraparib, and rucaparib. A total of 100 cases were considered, with a high prevalence (99.0%) in female sex. In terms of age group, 40 ICSRs (40.0%) do not refer to the age group, followed by 33 (33.0%) cases in elderly (64–85 years), and 24 (24.0%) cases in adults (18–64 years). Concerning the outcome, most cases reported a not recovered/not resolved outcome (N = 47, 47.0%) and all of them indicated the seriousness criteria "other medically important information". However, the high percentage of the cases reported as "unknown" should be highlighted (N = 44, 44.0%). Drug withdrawal was the most frequent action taken (N = 52, 52.0%). In addition, most ICSRs reported only one nervous ADRs (N = 49, 49.0%), considering 1295 overall (nervous and non-nervous) ADRs reported. For concomitant medicines, the majority of ICSRs do not contain any information in this field (N = 69, 69.0%). These results are mentioned in Table 3.

**Table 3.** Characteristics of individual cases of peripheral neuropathy reported from 1 January 2022 to 31 December 2022.

| | Individual Case Safety Reports (%) | | | |
|---|---|---|---|---|
| | Olaparib N = 2 | Niraparib N = 87 | Rucaparib N = 11 | Total N = 100 |
| Sex | | | | |
| Female | 2 (100.0) | 86 (98.9) | 11 (100.0) | 99 (99.0) |
| Not specified | 0 | 1 (1.1) | 0 | 1 (1.0) |
| Age group | | | | |
| Adult (18–64 years) | 2 (100.0) | 21 (24.1) | 1 (9.1) | 24 (24.0) |
| Elderly (65–85 years) | 0 | 24 (27.6) | 9 (81.8) | 33 (33.0) |
| Very Eldery (>85 years) | 0 | 3 (3.5) | 0 | 3 (3.0) |
| Not Specified | 0 | 39 (44.8) | 1 (9.1) | 40 (40.0) |
| Outcome | | | | |
| Recovered/Resolved | 0 | 3 (3.5) | 0 | 3 (3.0) |
| Recovering/Resolving | 0 | 5 (5.7) | 1 (9.1) | 6 (6.0) |
| Not recovered/Not resolved | 1 (50.0) | 42 (48.3) | 4 (36.4) | 47 (47.0) |
| Unknown | 1 (50.0) | 37 (42.5) | 6 (54.5) | 44 (44.0) |
| Seriousness Criteria | | | | |
| Other = other medically important information | 2 (100.0) | 87 (100) | 11 (100) | 100 (100.0) |
| Action Taken | | | | |
| Dose Reduced | 2 (100.0) | 15 (17.2) | 1 (9.1) | 18 (18.0) |
| Dose Increased | 0 | 4 (4.6) | 0 | 4 (4.0) |
| Drug withdrawn | 0 | 47 (54.0) | 5 (45.4) | 52 (52.0) |
| Dose not changed | 0 | 12 (13.8) | 4 (36.4) | 16 (16.0) |
| Unknown | 0 | 9 (10.5) | 1 (9.1) | 10 (10.0) |
| Number of nervous disorders per ICSR | | | | |
| 1 | 1 (50.0) | 42 (48.3) | 6 (54.5) | 49 (49.0) |
| 2 | 1 (50.0) | 27 (31.0) | 5 (45.5) | 33 (33.0) |

**Table 3.** *Cont.*

| | Individual Case Safety Reports (%) | | | |
|---|---|---|---|---|
| | Olaparib<br>N = 2 | Niraparib<br>N = 87 | Rucaparib<br>N = 11 | Total<br>N = 100 |
| 3 | 0 | 11 (12.7) | 0 | 11 (11.0) |
| 4 | 0 | 4 (4.6) | 0 | 4 (4.0) |
| 5 or more | 0 | 3 (3.4) | 0 | 3 (3.0) |
| Concomitant medicines per ICSR | | | | |
| 1 | 0 | 8 (9.2) | 0 | 8 (8.0) |
| 2 | 0 | 3 (3.4) | 0 | 3 (3.0) |
| 3 | 0 | 0 | 0 | 0 |
| 4 | 0 | 3 (3.4) | 0 | 3 (3.0) |
| 5 or more | 0 | 11 (12.7) | 6 (54.5) | 17 (17.0) |
| Not reported | 2 (100.0) | 62 (71.3) | 5 (45.5) | 69 (69.0) |
| Total suspected ADRs reported | | | | |
| Total number | 9 | 1134 | 152 | 1295 |
| Median per ICRS | 4.5 | 11 | 10 | 10 |

## 4. Discussion

PARPi are a novel class of targeted cancer therapies that have shown promising results in various types of oncological pathologies such as ovarian, breast, prostate, and pancreatic cancers [1,39]. Additionally, the literature has highlighted several associations, particularly the combination of these agents with immunotherapy and chemotherapy [31,40,41]. This study intended to investigate spontaneous reports related to the approved PARPi, olaparib, rucaparib, niraparib, and talazoparib, through the analysis of data obtained from EV, to provide an overview of suspected ADRs, with a focus on nervous disorders, particularly peripheral neuropathy.

A total of 12,762 ICSRs were retrieved from the date of marketing authorization granted by EMA to 1 March 2023 for each PARPi. Female patients are the most reported cases by healthcare professionals, especially in the age groups 18–64 years and 65–85 years. This fact can be explained through the approved therapeutic indications for this class of drugs. Olaparib and niraparib were the first two approved PARPi in the European Union market, approved in 2014 and 2017, respectively, and have provided great clinical benefits to ovarian cancer patients [19,42]. In addition, rucaparib was approved in 2018 by EMA and has a beneficial role in this type of cancer [24,43]. In a general way, "General disorders and administration site conditions" is the most reported SOC among PARPi with 4468 ICSRs analyzed. These results are aligned with what has already been described in the Summary of Product Characteristics (SmPC) and by some authors. Fatigue, regardless of the grade, is considered a very common symptom [20–23] in patients taking these drugs, representing a percentage about 59–69% of patients who experienced it [9,16,24]. PARP1, the main enzymatic target of PARPi, has been implicated in the regulation of circadian metabolic activities and it can be hypothesized that the disruption of these metabolic activities could be the basis for the high frequency of fatigue [44,45]. LaFargue et al. also pointed out investigational toxicities (hypercholesterolemia and increased amounts of serum hepatic enzymes), as well as gastrointestinal and haematological toxicities, as frequent among PARPi [31]. The inhibition of PARP2 may be involved in the development of haematological toxicity as PARP2 has been shown to have a role in the regulation of red blood cell production [44,45]. In addition, nervous, respiratory, psychiatric and skin disorders are considered less common toxicities when compared to the others described [31].

Considering the typology of reported ADRs among PARPi, in more serious cases, it was possible to observe a greater number of ICSRs in patients taking niraparib. Through a detailed analysis, differences in terms of ADRs frequency can be described [46]. For example, in terms of respiratory symptoms, dyspnoea and cough were reported in 79 and 37 ICSRs, respectively. However, interstitial lung disease was reported in 43 ICSRs associated with olaparib [47,48]. It should be noted that this adverse effect is not described in olaparib's SmPC [20]. Some cases of suspected interstitial lung disease have also been described for niraparib, which also does not have this adverse effect described in the SmPC [22]. According to the literature, the mechanism of respiratory toxicities is not well defined. However, preclinical data have described that the activation of PARP enzymes is associated with bronchial hyper-reactivity and airway remodeling [49]. In terms of psychiatric disorders, 155 serious cases of insomnia were reported with the niraparib inhibitor. According to the SmPC, insomnia is considered a very common ADR in patients taking this drug. Additionally, anxiety and depression are considered common reactions [22]. The disparity between niraparib vs. olaparib/rucaparib in terms of psychiatric ADRs may be related to niraparib's pan-neurotransmitter pharmacology [50]. Additionally, headache and dizziness were also mainly reported in patients taking niraparib. These results have already been highlighted in the literature [9,16,24]. However, a definitive link could not be established due to incomplete reporting of all off-target profiles of PARPi [50]. Concerning the suspected ADRs peripheral neuropathy, 87 serious ICSRs were found, in 2022, for niraparib, followed by 11 cases in patients taking rucaparib and 2 cases associated with olaparib. This ADRs is not mentioned in niraparib, rucaparib, or olaparib SmPC's [20–22].

The characteristics of those ICSRs that reported peripheral neuropathy were analyzed in more detail for the inhibitors olaparib, niraparib, and rucaparib. The majority of cases were reported in female patients (99.0%) and in adults (18–64 years) and elderly (65–85 years) people (57.0%). Regarding the outcome, although 44.0% of the cases did not contain any information, it was possible to observe that 47.0% of the cases were classified as a not recovered/not resolved outcome. Furthermore, all of them were classified with the seriousness criteria "Other medically important information". Drug withdrawal was the action applied to about half of the cases (52.0%), followed by dose reduction (18.0%). Guo et al. recently conducted a real-world pharmacovigilance study based on suspected ADRs for niraparib reported to the FAERS that described peripheral neuropathy as an unexpected significant ADRs. In this study, 649 cases were reported between 2017 and 2021 [32]. Another recent study using the FAERS database mentioned 362 reports of peripheral neuropathy. In this context, Tian et al. mentioned that the reporting odds ratio (ROR) in signals detections suggested that neurotoxicity might be more frequent in patients treated with niraparib [47]. According to these authors, further investigations are needed in this area. Additionally, mostly preclinical studies suggested that, when compared with conventional chemotherapy, PARPi may help with symptoms related to peripheral neuropathy [51–54]. However, a meta-analysis performed by Balko et al. showed that PARP inhibition activity does not appear to reduce the risk of developing neuropathy induced by chemotherapy [55]. Considering the above, new pharmacovigilance studies should be conducted to clarify more precisely the toxicity profile of these drugs.

For this study, some strengths and limitations should be considered. The major strength was the access to a large and comprehensive spontaneous reports database on PARPi. We were able to analyze ICSRs from a heterogeneous population, which is usually not considered in the premarketing clinical trials. Despite our best efforts to conduct a sound study and to minimize bias, there are important limitations that need to be acknowledged, namely the lack of crucial information in ICSR, such as outcome, action taken and its results, and concomitant medicines, among others. The phenomenon of underreporting and underestimation of the frequency of ADRs in oncology is also a major problem that needs to be considered [56]. Additionally, this data cannot provide evidence on the causal relationship between ADRs and the suspected drugs.

## 5. Conclusions

A retrospective analysis was conducted on data retrieved from the EV database. The spontaneous reporting system represents a useful tool for understanding drug safety data and for better characterization of drug safety profiles. Although PARPi are targeted drugs, several systems/organs may be affected by taking these drugs. Unexpected ADRs may also occur, such as peripheral neuropathy. More high-quality pharmacovigilance studies should be conducted to better understand the PARPi toxicity profile. In this context, new guidelines for healthcare professionals to manage these adverse effects in a real setting may be provided.

**Author Contributions:** Conceptualization, M.J. and M.M.; methodology, M.J. and A.C.; validation, M.J. and M.M.; formal analysis, M.J., A.C., C.M., A.P.D. and M.M.; investigation, M.J., A.C. and M.M.; writing—original draft preparation, M.J.; writing—review and editing, A.C., C.M., A.P.D. and M.M.; supervision, C.M., A.P.D. and M.M.; project administration, A.P.D. and M.M. All authors have read and agreed to the published version of the manuscript.

**Funding:** This work was partially supported by CICS-UBI, which is financed by national funds from Fundação para a Ciência e a Tecnologia (FCT) and by Fundo Europeu de Desenvolvimento Regional (FEDER) under the scope of PORTUGAL 2020 and Programa Operacional do Centro (CENTRO 2020), with the project reference numbers UIDB/00709/2020 and UIDP/00709/2020.

**Institutional Review Board Statement:** Not applicable.

**Informed Consent Statement:** Not applicable.

**Data Availability Statement:** All data that the authors extracted from the European Medicines Agency (EMA) pharmacovigilance database called EudraVigilance are publicly available. Pharmacovigilance data from EudraVigilance is publicly available at www.adrreports.eu (accessed on 8 March 2023).

**Conflicts of Interest:** The authors declare no conflict of interest.

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
