# Peer review of "Peripheral Neuropathy Potentially Associated to Poly (ADP-Ribose) Polymerase Inhibitors: An Analysis of the Eudravigilance Database"

_curroncol, doi:10.3390/curroncol30070479_

Round 1
Reviewer 1 Report
Comments and Suggestions for Authors:
By analyzing the individual case safety reports retrieved from EV database, the author described the adverse drug reactions caused by the use of PARP inhibitors, which would help to clarify the toxicity characteristics of the PARPi more accurately in the future. It is an important study, but there are some problems, which must be solved before it is considered for publication.
Some points are required:
1. The presentation of the results is not very clear, adding some pie charts may be helpful.
2. In the discussion section, the author can discuss the potential mechanisms of adverse drug reactions caused by the use of PARPi.
3. Authors need to check the spellings in current manuscript.
English language fine. No issues detected
Author Response
Responses to Reviewers
The authors are grateful for comments and suggestions from all reviewers and the Editor. All comments and suggestions have been taken into account, which has helped to improve the quality of the article. All changes to the article are marked in yellow.
The English has also been revised by a native English teacher. All these English corrections have not been reported so as not to unnecessarily burden the revised article with corrections.
Point-to-point responses to Reviewer 1's comments
- The presentation of the results is not very clear, adding some pie charts may be helpful.
The authors are grateful for this comment and suggestion and have taken the opportunity to add 5 pie charts in the “Results” section, which contributed to improve the quality of the article.
These pie charts were included in Figure 1, which presents the contribution of each studied category (between gender, age group, reporter group, region, and seriousness) to the total of ICSRs analysed (see Figure 1, lines 175-178 of the revised manuscript).
Likewise, and with a view to clarifying the presentation of results presented in “3.2. Least reported SOCs”, the authors prepared Figure 2, which describes the distribution of the total serious ICSRs reported for each PARPi versus serious ICSRs reported for some selected SOCs (the “Least reported SOCs” of 3.2.), only in the year of 2022 (from 1 January 2022 to 31 December 2022). (see Figure 2, lines 203-205 of the revised manuscript).
The authors also sought to better explain that for each SOC included in Table 2, only the most reported safety problems were included, considering all PARPi. (see lines 186-192 of the revised manuscript; see also the revised legend of Table 2 in lines 196-198).
- In the discussion section, the author can discuss the potential mechanisms of adverse drug reactions caused by the use of PARPi.
The authors agree with this comment and suggestion which contributed to improve the quality of the article. Considering the Reviewer's suggestion, the authors added the following sentences in “Discussion”:
PARP1, the main enzymatic target of PARPi, has been implicated in the regulation of circadian metabolic activities, and it can be hypothesized that the disruption of these metabolic activities could be the basis for the high frequency of fatigue [45,46]. (see lines 246-249 of the revised manuscript).
The inhibition of PARP2 may be involved in the development of haematological toxicity as PARP2 has been shown to have a role in the regulation of red blood cell production [44,45]. (see lines 252-254 of the revised manuscript).
- Authors need to check the spellings in current manuscript. Minor editing of English language required.
According to the Reviewers suggestions, the English of the whole article has been revised by a native English teacher, which contributed to improve the quality of the manuscript.

Reviewer 2 Report
The manuscript "peripheral neuropathy potentially associated to PARP inhibitors: An analysis from the Eudravigilance database" is an interesting and important study, and should definitely be published. However, unfortunately the English is unclear in places (especially in the introduction) which needs to be corrected before acceptance. For example in the introduction the text states:
"In this context, two of the most described methods relies on inhibition the PARP enzyme and on the “PARP trapping” concept, being PARP1 the main target [2,3]. PARP1 is considered the most abundant isoform and is recognized by its role in DNA repair processes [4]. Other isoforms such as PARP2, PARP3 and PARP5 are described in the literature [5–7]."
This language needs to be improved as it is hard to discern what is being said. Instead it should read something like:
"In this context, two of the most described methods relies on inhibition of the PARP enzyme and on the “PARP trapping” concept, being PARP1 the main target [2,3]."
OR
"In this context, two of the most described methods relies on inhibiting the PARP enzyme and on the “PARP trapping” concept, being PARP1 the main target [2,3]."
This editing should be completed on the whole manuscript, preferably with a native English speaker, so that the important results can be described effectively.
The English needs to be reviewed somewhat extensively to improve clarity, as per comments.
Author Response
Responses to Reviewers
The authors are grateful for comments and suggestions from all reviewers and the Editor. All comments and suggestions have been taken into account, which has helped to improve the quality of the article. All changes to the article are marked in yellow.
The English has also been revised by a native English teacher. All these English corrections have not been reported so as not to unnecessarily burden the revised article with corrections.
Point-to-point responses to Reviewer 2's comments
- However, unfortunately the English is unclear in places (especially in the introduction) which needs to be corrected before acceptance. For example in the introduction the text states:
"In this context, two of the most described methods relies on inhibition the PARP enzyme and on the “PARP trapping” concept, being PARP1 the main target [2,3]. PARP1 is considered the most abundant isoform and is recognized by its role in DNA repair processes [4]. Other isoforms such as PARP2, PARP3 and PARP5 are described in the literature [5–7]."
This language needs to be improved as it is hard to discern what is being said. Instead it should read something like:
"In this context, two of the most described methods relies on inhibition of the PARP enzyme and on the “PARP trapping” concept, being PARP1 the main target [2,3]."
OR
"In this context, two of the most described methods relies on inhibiting the PARP enzyme and on the “PARP trapping” concept, being PARP1 the main target [2,3]."
The authors are grateful for this comment and correction and have rewritten the mentioned sentence according to the first suggestion of Reviewer 2. (see lines 31-32 of the revised manuscript).
The English of the whole manuscript has also been revised by a native English teacher. All these English corrections have not been reported so as not to unnecessarily burden the revised article with corrections.
- This editing should be completed on the whole manuscript, preferably with a native English speaker, so that the important results can be described effectively.
Review Report Form: The English needs to be reviewed somewhat extensively to improve clarity, as per comments.
According to the Reviewers suggestions, the English of the whole article has also been revised by a native English teacher, which contributed to improve the quality of the manuscript.
- Review Report Form: Does the introduction provide sufficient background and include all relevant references? Must be improved.
The authors are grateful for this comment and suggestion and have taken the opportunity to provide additional background information in the “Introduction” section and to include some additional relevant references.
The following relevant background information and references has been added to the “Introduction” section of the manuscript:
According to the literature, 59-69% of patients had fatigue of any degree with olaparib, rucaparib and niraparib. [9, 16, 24] (references 9, 16 and 24 were added to this sentence; see lines 57-58 of the revised manuscript).
In the phase III trials [the SOLO-1 trial (olaparib monotherapy 300 mg twice daily versus placebo), the PAOLA-1 trial (olaparib 300 mg twice daily plus bevacizumab versus bevacizumab monotherapy) and the PRIMA trial (niraparib monotherapy 300 mg qd vs. placebo)], all PARPi as FDA-approved maintenance therapy after first-line platinum therapy showed high rates of a number of common non-hematologic ADR, such as nausea, vomiting, and fatigue/asthenia, which are generally low-grade and rarely lead to study drug discontinuation [25-27]. (this background information and these references were added; see lines 59-65 of the revised manuscript).
Among hematological adverse events, anemia is considered the most common. Grade 3 and 4 events are considered the most common cause of dose adjustment or drug discontinuation, particularly in patients taking niraparib [28]. Also, nausea and, to a lesser extent, vomiting are very common ADR associated with PARPi, especially in patients with ovarian cancer. Supportive treatment including antiemetics are usually effective, avoiding dose interruption and dose reduction actions [29,30]. Elevated creatinine, liver enzymes and cholesterol were also mentioned as common investigational toxicities of patients using PARPi [31]. (such additional information and references have been added; see lines 65-72 of the revised manuscript)
Dysgeusia is more common with olaparib and rucaparib, with cardiac/ cardiovascular disorders, particularly palpitations and hypertension, being more frequently associated with niraparib [19]. (this basic information and this reference have been added; see lines 75-77 of the revised manuscript).
Rare and delayed adverse effects should also be highlighted such as myelodysplastic syndrome and acute myeloid leukemia [19]. (such additional information and reference have been added; see lines 79-81 of the revised manuscript).
